# The Role of Protease-Activated Receptor 2 in Hepatocellular Carcinoma after Hepatectomy

**DOI:** 10.3390/medicina57060574

**Published:** 2021-06-04

**Authors:** Ming-Chao Tsai, Chih-Che Lin, Ding-Wei Chen, Yueh-Wei Liu, Yi-Ju Wu, Yi-Hao Yen, Pao-Yuan Huang, Chih-Chien Yao, Ching-Hui Chuang, Chang-Chun Hsiao

**Affiliations:** 1Division of Hepato-Gastroenterology, Department of Internal Medicine, Kaohsiung Chang Gung Memorial Hospital and Chang Gung University College of Medicine, Kaohsiung 83301, Taiwan; tony0779@gmail.com (M.-C.T.); cassellyen@yahoo.com.tw (Y.-H.Y.); paoyuan813@gmail.com (P.-Y.H.); chihchienyao@gmail.com (C.-C.Y.); 2Graduate Institute of Clinical Medical Sciences, Chang Gung University College of Medicine, Kaohsiung 83301, Taiwan; 3Liver Transplantation Center and Department of Surgery, Kaohsiung Chang Gung Memorial Hospital and Chang Gung University College of Medicine, Kaohsiung 83301, Taiwan; immunologylin@gmail.com (C.-C.L.); anthony0612@me.com (Y.-W.L.); wuyiju0904@gmail.com (Y.-J.W.); 4Center for Translational Research in Biomedical Sciences, Liver Transplantation Program and Department of Surgery, Kaohsiung Chang Gung Memorial Hospital and Chang Gung University College of Medicine, Kaohsiung 83301, Taiwan; dennis8870@gmail.com; 5Department of Nursing, Meiho University, Pingtung 91202, Taiwan; Helen.ch.chuang@gmail.com; 6Graduate Institute of Clinical Medical Sciences, College of Medicine, Chang Gung University, Division of Pulmonary and Critical Care Medicine, Kaohsiung Chang Gung Memorial Hospital, 123, Ta-Pei Road, Niao-Sung District, Kaohsiung 833, Taiwan

**Keywords:** PAR2, hepatocellular carcinoma, liver resection, AFP

## Abstract

*Background and Objectives*: Protease activated receptor-2 (PAR2) is elevated in a variety of cancers and has been promoted as a potential therapeutic target. However, the clinical and prognostic values of PAR2 in hepatocellular carcinoma (HCC) are poorly characterized. This study aimed to evaluate the expression of PAR2 in HCC tissues and examine the prognostic value of PAR2 after resection in HCC. *Materials and Methods*: Two hundred and eight resected specimens were collected from HCC patients at Kaohsiung Chang Gung Memorial Hospital. PAR2 protein expression was assessed by western blotting in HCC tissues and matched normal tissues. The correlation between PAR2 expression and clinicopathological parameters was analyzed. Disease-free survival (DFS) and overall survival (OS) were compared using the log-rank test. A Cox regression model was used to identify independent prognostic factors. *Results*: PAR2 was expressed at higher levels in HCC tissues than the paired adjacent nontumor tissues. High expression of PAR2 was associated with advanced tumor, node, metastasis (TNM )stage and histological grade. Kaplan-Meier analysis indicated high PAR2 expression was associated with poorer DFS and OS compared to low PAR2 expression. Multivariate analyses indicated high PAR2 expression [hazard ratio (HR), 1.779, *p* = 0.006), α-fetoprotein (AFP) (HR, 1.696, *p* = 0.003), liver cirrhosis (HR, 1.735, *p* = 0.002), and advanced TNM stage (HR, 2.061, *p* < 0.001) were prognostic factors for DFS, and advanced TNM stage (HR, 2.741, *p* < 0.001) and histological grade (HR, 2.675, *p* = 0.002) and high PAR2 expression (HR, 1.832, *p* = 0.012) were significant risk factors for OS. In subgroup analyses, the combination of PAR2 expression and serum AFP provided improved prognostic ability for OS and DFS. *Conclusion*: Combination PAR2 and AFP predict HCC outcomes after resection. PAR2 represents a potentially clinically relevant biomarker for HCC.

## 1. Introduction

Hepatocellular carcinoma (HCC), the sixth most common tumor type, is a major cause of cancer-related deaths around the world [1,2]. Although a number of therapeutic options exist, including liver transplantation, hepatectomy, and ablation, overall survival is still poor due to the high rate of recurrence (59–60%) [3]. Several factors are prognostic for recurrence and/or survival after resection in HCC, including tumor size and differentiation, serum α-fetoprotein (AFP), microvascular invasion, cirrhosis, surgical margin and metabolic syndrome [4,5,6]. However, more effective prognostic biomarkers need to be identified to improve the prediction of outcomes in HCC. The ability to identify patients at high risk of recurrence would enable clinicians to provide more intensive surveillance and detect recurrence at an earlier stage, when curative therapy may still be possible.

Protease-activated receptors (PARs) is a G-protein-coupled receptor that is activated by proteolytic cleavage of their extracellular N terminal domain [7]. It is also activated by trypsin, mast cell tryptase, and the tissue factor/factor VIIa and factor Xa complex [8]. A couple of studies demonstrated that PARs were associated with the modulation of vascular, inflammatory response, fibrogenesis, and carcinogenesis, which made PARs the potential targets for innovative therapies development [9,10,11]. In the setting of liver cancer, it could be demonstrated that PAR2 is expressed in HCC tissues, and in HCC cell lines, PAR2 can stimulate cell migration and invasion through different signaling pathways [12,13]. Hence, a crucial role of PAR2 in HCC progression can be hypothesized. However, the expression, function and clinical value of PAR2 in HCC have not been investigated. This study was designed to determine the expression of PAR2 in HCC tissues and examine the prognostic value of PAR2 after resection in HCC.

## 2. Materials and Methods

### 2.1. Ethics

This study was performed in accordance with the Declaration of Helsinki and approved by the Institutional Review Board of Chang Gung Memorial Hospital (IRB number: 201800049B0; Date: from August 2018 to July 2019). The ethics committee waived the requirement for informed consent; all patient data were anonymized.

### 2.2. Tissue Specimens and Clinical Data

Samples from 208 patients with HCC who underwent curative hepatic resection at Kaohsiung Chang Gung Memorial Hospital, Taiwan, between January 2008 and December 2015, were obtained from Tissue Bank, Kaohsiung Chang Gung Memorial Hospital. The samples included 208 paired HCC tumor/non-tumor tissues.

The resected tumors were histologically diagnosed according to international guidelines [14,15,16,17,18]. Disease-free survival (DFS) was calculated from the day of surgery until detection of recurrent or metastatic HCC by liver computed tomography (CT) or magnetic resonance imaging (MRI); overall survival (OS), from surgery to death, last contact, or December 2018. Survival outcomes were assessed using the patients’ final medical records; data on patient demographic and clinical characteristics were also extracted from medical records, including age, gender, clinical stage (defined by the American Joint committee on Cancer tumor, node, metastasis (TNM) staging system [19]), tumor differentiation (according to the Edmondson-Steiner system [20]), number of tumors, maximal tumor size (according to the postoperative pathology), pathological microvascular invasion, serum hepatitis markers (HBsAg and anti-HCV), serum alpha-fetoprotein (AFP) level, and cirrhosis (Ishak score of 5 or 6 [21]).

### 2.3. Western Blotting

Tissue samples (50 mg) were homogenized in lysis buffer (40 mmol/L *N*-2-hydroxyethyl-piperazine *N*′-2-ethane sulphonate, 1% Triton X-100, 10% glycerol, 1 mmol/L phenylmethanesulfonyl fluoride), sonicated, centrifuged at 12,000 rpm for 30 min at 4 °C and the total protein extracts (25–50 μg/lane) were subjected to immunoblot analysis. Proteins were transferred onto polyvinylidene di-fluoride membranes (Immobilon-P membrane; Millipore, Bedford, MA, USA), incubated with anti-PAR2 antibody (ab180953, Abcam, Cambridge, UK) overnight at 4 °C, washed, incubated with anti-rabbit and anti-mouse IgG secondary peroxidase-conjugated antibodies at 25 °C for 1 h (Cell Signaling, Danvers, MA, USA), and the bands were developed using the ECL plus chemiluminescence kit (Amersham Pharmacia Biotech, Buckinghamshire, UK). Target protein expression values were normalized to β-actin. Two-fold higher PAR2 protein expression in the tumor sample compared to the matched nontumor tissue was defined as high PAR2 expression.

### 2.4. Statistical Analyses

Data were analyzed using SPSS version 22 (IBM, Chicago, IL, USA) and expressed as median (interquartile range, IQR) or mean ± standard deviation. Categorical data were compared using the chi-square test. DFS and OS were assessed using the Kaplan-Meier method and compared using the log-rank test. The Cox proportional hazards model was used to identify independent risk factors for DFS and OS; factors with a *p*-value < 0.3 in univariate analysis were included in the multivariate analysis. Two-sided *p*-values <0.05 were considered significant.

## 3. Results

### 3.1. Patient Characteristics

The clinicopathological features of the 208 patients with HCC who underwent curative resection are summarized in Table 1. Overall, the cohort included 162 males (78%) and 46 females (22%), with a mean age of 59 years (range: 25–82). The etiology of HCC was associated with hepatitis B virus (HBV) in 120 cases (59.1%) and hepatitis C virus (HCV) in 63 cases (31%); the remaining 25 cases (12%) had other/unknown etiologies. The median largest tumor diameter was 5 cm (IQR 3.5–8 cm). Pathological examination confirmed microvascular invasion in 105 cases (53.8%). Mean follow-up duration was 58 months (range, 2–140 months). Of the 142 (68.3%) patients who developed recurrence, the mean time to recurrence was 40.1 months. Overall, 112 patients (53.8%) died during follow-up.

### 3.2. Association between PAR2 and the Clinicopathological Features of HCC

The expressions of PAR2 were examined by western blot analysis in 208 paired HCC and nontumor tissues (Figure 1). Compared with the paired nontumor tissues, high levels (defined as greater than twofold increase) of PAR2 expression in 143 of 208 (68.8%) HCC cases, and the other 65 (31.2%) were defined as PAR2 low expression group. The clinicopathological features of patients with HCC stratified by PAR2 expression are summarized in Table 2. We found that patients with high expression of PAR2 were associated with advanced tumor-node-metastasis (TNM) stage (*p* = 0.005), and relative high serum AFP level (*p* = 0.060), but not with other characteristics such as gender, age, etiology, tumor, size, vascular invasion, and pathological stage.

### 3.3. Association between PAR2 and Recurrence in HCC

After the median follow-up of 24 months (IQR, 6.3–60.3), 142/208 (68.3%) patients had developed recurrence. The cumulative incidence of recurrence for patients with high PAR2 expression was 39.7% at year 1, 65.3% at year 3, and 72.7% at year 5. In contrast, the cumulative incidence of recurrence was 22.0% at year 1, 35.1% at year 3, and 55.5% at year 5 for patients with low PAR2 expression (*p* < 0.001; Table 3 and Figure 2A). In addition to PAR2 expression, serum AFP (≥200 ng/mL; *p* < 0.001), cirrhosis (*p* = 0.001), tumor size (≥5 cm; *p* = 0.001), number of tumors (multiple tumors; *p* = 0.006), vascular invasion (*p* < 0.001), advanced TNM stage (*p* < 0.001), and advanced histological grade (*p* < 0.001) were associated with recurrence in HCC (Table 3).

Univariate analyses using the Cox proportional hazard model demonstrated that AFP ≥ 200 ng/mL (hazard ratio (HR), 1.675; 95% CI, 1.284–2.385; *p* < 0.001), liver cirrhosis (HR, 1.647; 95% CI, 1.204–2.253; *p* = 0.002), tumor size ≥ 5 cm (HR, 1.668; 95% CI, 1.228–2.266; *p* = 0.001), multiple tumors (HR, 1.603; 95% CI, 1.14–2.254; *p* = 0.007), vascular invasion (HR, 2.625; 95% CI, 1.902–3.622; *p* < 0.001), TNM stage (III/IV vs. I/II; HR, 2.655; 95% CI, 1.95–3.613; *p* < 0.001), pathological grade (poor/moderate vs. well; HR, 1.95; 95% CI, 1.304–2.916; *p* = 0.001), and PAR2 expression (high vs. low; HR, 1.969; 95% CI, 1.144–3.388; *p* = 0.014) were associated with a significantly higher risk of recurrence. These significant covariates from univariate analysis were entered into multivariate Cox analysis. Serum AFP ≥ 200 ng/mL (HR, 1.696; 95% CI, 1.196–2.403; *p* = 0.003), liver cirrhosis (HR, 1.735; 95% CI, 1.221–2.466; *p* = 0.002), TNM stage (III/IV vs. I/II; HR, 2.061; 95% CI, 1.447–2.934; *p* < 0.001), and PAR2 expression (high vs. low; HR, 1.779; 95% CI, 1.181–2.681; *p* = 0.006) were identified as independent risk factors for recurrence in HCC (Table 4).

### 3.4. Association between PAR2 and Overall Survival

After the median follow-up of 57 months (IQR, 19–84), 112/208 (53.8%) patients had died. The OS rates of patients with high PAR2 expression were 78.8% at year 1, 56.9% at year 3, and 49.9% at year 5 compared with 95.2% at year 1, 80.9% at year 3, and 67.3% at year 5 for patients with low PAR2 expression (*p* = 0.002; Table 5 and Figure 2B). In addition to Chibby, serum AFP (≥200 ng/mL; *p* = 0.002), cirrhosis (*p* < 0.001), vascular invasion (*p* < 0.001), advanced TNM stage (*p* < 0.001), and advanced histological grade (*p* < 0.001) were associated with poorer OS after resection in patients with HCC.

Univariate analyses indicated serum AFP ≥ 200 ng/mL (HR, 1.721; 95% CI, 1.221–2.426; *p* = 0.002), tumor size ≥ 5 cm (HR, 2.083; 95% CI, 1.469–2.955; *p* < 0.001), presence of microvascular invasion (HR, 3.231; 95% CI, 2.122–4.696; *p* < 0.001), advanced TNM stage (III/IV vs. I/II; HR, 3.356; 95% CI, 2.366–4.761; *p* < 0.001), pathological grade (poor/moderate vs. well; HR, 2.946; 95% CI, 1.719–5.605; *p* < 0.001), and PAR2 expression (high vs. low; HR, 2.027; 95% CI, 1.29–3.184; *p* = 0.002) were associated with significantly poorer OS. Multivariate Cox analysis of the significant covariates from univariate analyses revealed that TNM stage (III/IV vs. I/II; HR, 2.747; 95% CI, 1.851–4.077; *p* < 0.001), pathological stage (poor/moderate vs. well; HR, 2.675; 95% CI, 1.417–5.051; *p* = 0.002), and PAR2 expression (high vs. low; HR, 1.832; 95% CI, 1.142–2.938; *p* = 0.012) were significant independent prognostic factors for OS (Table 6).

### 3.5. Prognostic Value of Chibby Combined with Serum AFP

As serum AFP ≥ 200 ng/mL was prognostic for poorer DFS and OS, we explored whether the prognostic value of PAR2 varied with serum AFP. The patients were divided into four subgroups: high PAR2 and AFP ≥ 200 ng/mL (*n* = 59); high PAR2 and AFP < 200 ng/mL (*n* = 84); low PAR2 and AFP ≥ 200 ng/mL (*n* = 18); and low PAR2 and AFP < 200 ng/mL (*n* = 47). Patients with low PAR2 and AFP < 200 ng/mL had significantly better DFS (*p* < 0.001, Figure 3A) and OS (*p* < 0.001, Figure 3B) rates than all other subgroups. Among patients with serum AFP < 200 ng/mL, the low PAR2 subgroup achieved significantly better DFS (*p* = 0.024, Figure 3A) and OS (*p* = 0.058, Figure 3B) than the high PAR2 subgroup. Similarly, among patients with AFP ≥ 200 ng/mL, the low PAR2 subgroup had significantly better DFS (*p* = 0.007, Figure 3A) and OS (*p* = 0.014, Figure 3B) than the high PAR2 subgroup.

## 4. Discussion

Developments in surgical techniques and new targeted drugs have markedly improved the treatment of HCC in recent years [22]. However, outcomes remain poor due to the high rate of recurrence after resection. Although several biomarkers have prognostic value for outcome after surgery, more effective novel biomarkers and targets for HCC urgently need to be identified.

PAR2, a member of the G-protein coupled receptor 1 family [7], is expressed in a wide range of cellular types, where it has been involved in multiple physiological and pathophysiological processes including cancers [23,24]. Although PAR2 has been implicated in the pathogenesis of several cancers, including lung, skin, stomach, ovary, kidney and colon [25,26,27,28,29], and mounting in vitro studies suggested a role for PAR2 in cancer development, including cell proliferation, invasion, and metastasis [30,31], only a limited number of studies of PAR2 in the HCC are currently available. In this study, we used western blot analysis to confirm that PAR2 is upregulated in the majority of HCC tissues (up to 70%). Moreover, upregulation of PAR2 was significantly associated with advanced TNM stage and poor differentiation, and PAR2 tumor expression was an independent prognostic factor for OS and DFS after resection. To our best of our knowledge, this is the largest cohort to evaluate the expression of PAR2 in predicting the outcomes of HCC after resection. Therefore, we confirm that assessment of PAR2 expression could potentially help to predict the outcome of patients with HCC; patients with high expression of PAR2 may benefit from more intensive surveillance and timely adjuvant treatment to improve their prognosis.

Our prior studies have demonstrated that coagulants tissue factor (TF) and factor VII (FVII) has a pathological role in promoting hepatoma growth by activating PAR2 [32,33]. We found that activation of TF/FVII/PAR2 axis is associated with increased invasiveness and migration in HCC cell lines in vitro, which is mainly via extracellular signal-regulated kinase-tuberous sclerosis complex (ERK-TSC). This mechanism could be explained by the increased expression of PAR2 in HCC tissues were associated with advanced TNM stage and poor DFS and OS. These findings were consistently observed in the study by Chen, in which PAR2 expression was increased in HBV-related HCC and high PAR2 expression was correlated with both poor DFS and OS [34].

In addition to PAR2, we found that serum AFP, cirrhosis, and advanced TNM stage were independent risk factors for recurrence and advanced TNM and histological grade were independent risk factors for OS. These data are in agreement with reports that tumor-related factors may determine the outcomes of patients with HCC after resection [35,36]. AFP is a valuable, cost-effective serum biomarker used to assess the prognosis of AFP-positive patients with HCC in clinical practice. To explore the impact of serum AFP on DFS and OS in more detail, we performed a subgroup analysis based on AFP and PAR2. Patients with high expression of PAR2 and high serum AFP levels had significantly poorer DFS and OS rates than the other subgroups. Thus, the combination of AFP with PAR2 may provide an accurate prognostic tool for HCC. It is worth noting that low expression of PAR2 identified patients with a good prognosis in terms of DFS and OS in both the high and low serum AFP subgroups. Future studies are required to validate the clinical value of PAR2 in patients with HCC undergoing other treatments, such as radiofrequency ablation, transcatheter arterial chemoembolization, target therapy, and immunotherapy.

Collectively, we propose that PAR2 levels in HCC tissues may provide prognostic value, which may be particularly useful in the prediction of HCC patients after resection. It might be practical to provide adjuvant therapy for HCC patients with higher PAR2 expression after resection, who had poorer DFS and OS rate. A recent study suggests that I-191 may be a valuable antagonist of human PAR2 in cancer cells [37]. Thus, further clinical trials with adjuvant therapy after HCC resection are needed to verify the prognostic efficacy of PAR2.

Our study had some potential limitations. First, this is a retrospective study with patients from a single tertiary medical center, some patients didn’t return to our hospital for further follow-up or even died after the operation, which could lead to biases. Second, more than half of the patients in the study are HBV infection, which is different from Western countries. However, we believe this difference did not affect our result due to no significant differences of PAR2 expression between hepatitis B and C. Finally, further studies in health individuals, and patients with other liver diseases, such as alcoholic liver disease and metabolic associated fatty liver disease, should be done to validate the role of PAR2 in the general population.

## 5. Conclusions

In conclusion, this study confirms that PAR2 is upregulated in HCC and has prognostic value after resection of HCC. Combination of PAR2 and AFP could represent a potentially useful prognostic factor in HCC and pave the way for the identification of novel pharmacological agents that target PAR2.

## Figures and Tables

**Figure 1 medicina-57-00574-f001:**
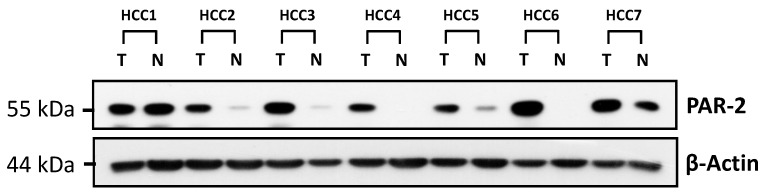
Western blot analysis of PAR2 in seven representative HCC tissue (T) and their paired non-tumor (N) tissue. β-Actin was used as a loading control.

**Figure 2 medicina-57-00574-f002:**
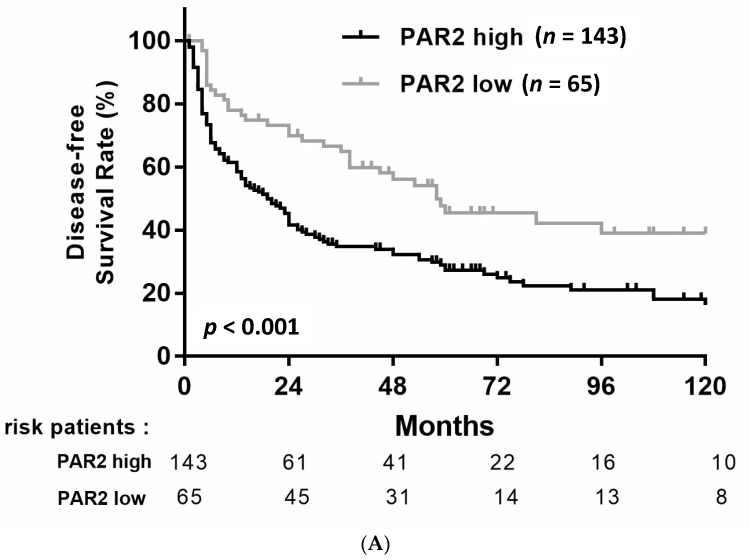
Disease-free survival (**A**) and overall survival (**B**) of HCC patients after resection stratified by PAR2 expression.

**Figure 3 medicina-57-00574-f003:**
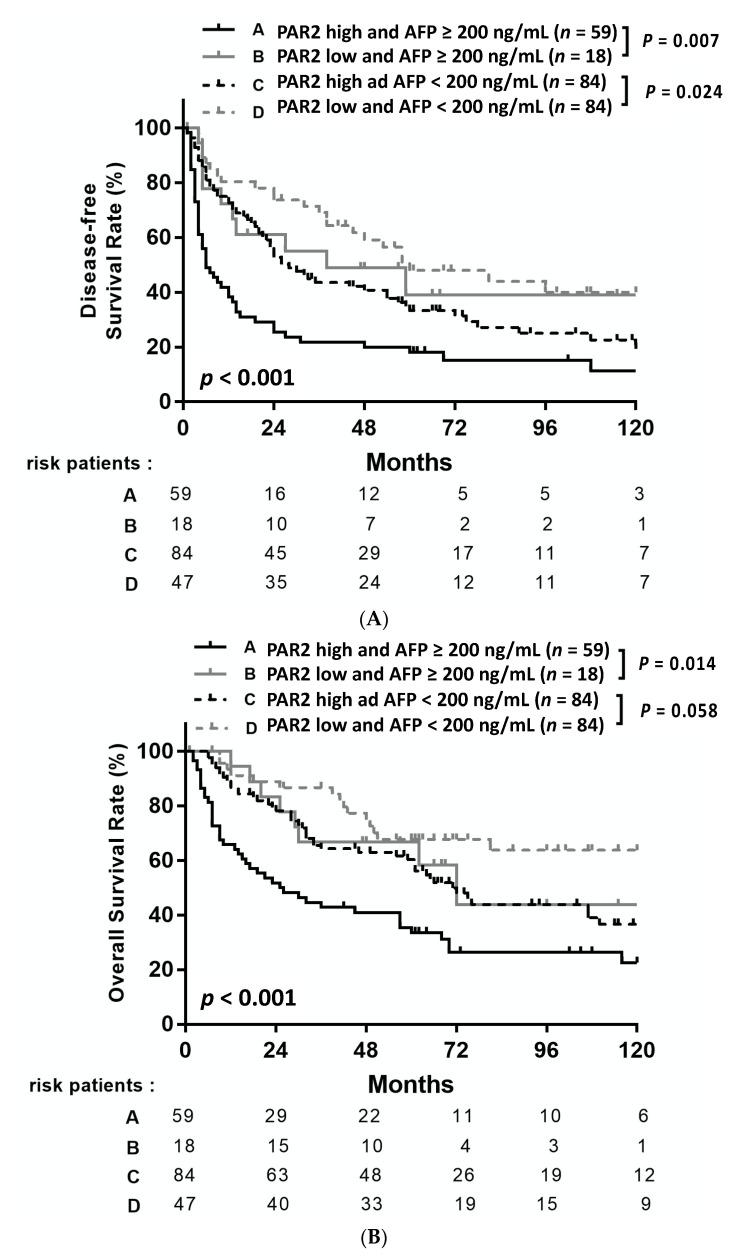
Disease-free survival (**A**) and overall survival (**B**) of HCC patients after resection stratified by the combination of PAR2 expression and AFP levels.

**Table 1 medicina-57-00574-t001:** Clinicopathological features of the 208 patients with HCC undergoing tumor resection.

**Clinicopathological Feature**	
Age [years, mean (SD)]	59 (13.5)
Male gender, *n* (%)	162 (78)
AFP [ng/mL median (IRQ)]	48 (5–825)
AFP > 15 ng/mL, *n* (%)	124 (59.6)
AFP > 200 ng/mL, *n* (%)	77 (37)
Liver cirrhosis, *n* (%)	108 (51.9)
Etiology	
Hepatitis B	120 (59.1)
Hepatitis C	63 (31)
Other/unknown	25 (12)
**Tumor Characteristics**	
Tumor size [cm, median (IQR)] ^a^	5 (3.5–8)
Tumor size > 3 cm, *n* (%)	169 (81.3)
Tumor size > 5 cm, *n* (%)	103 (49.5)
Solitary tumor, *n* (%)	166 (79.8)
Microvascular invasion, *n* (%)	105 (53.8)
TNM stage (I:II:III:IV)	44:78:54:31
Histological grade (well:moderate:poor)	
Well	41 (19.7)
Moderate	130 (62.5)
Poor	37 (17.8)
**Clinical Outcome**	
HCC recurrence, *n* (%)	142 (68.3)
Mean time to recurrence, [months, mean (SD)]	40.1 (43.1)
Died, *n* (%)	112 (53.8)
Mean time to death, [months, mean (SD)]	58 (44.6)

Data are expressed as median (interquartile range), mean (standard deviation) or number (percentage). ^a^ Diameter of the largest tumor nodule. AFP, alpha-fetoprotein; IQR, interquartile range; SD, standard deviation, TNM, tumor node metastasis.

**Table 2 medicina-57-00574-t002:** Association between PAR2 and the clinicopathological features of HCC.

Variable	PAR2 Expression (*n* = 208)		*p* Value
	Low (*n* = 65)	High (*n* = 143)	
Gender			0.096
Female	19 (29.2)	27 (18.9)	
Male	46 (70.8)	116 (81.1)	
Age (years)			0.155
<60	29 (44.6)	79 (55.2)	
≥60	36 (55.4)	64 (44.8)	
AFP (ng/mL)			0.060
<200	47 (72.3)	84 (58.7)	
≥200	18 (27.7)	59 (41.3)	
HBsAg			0.149
Negative	30 (48.4)	53 (37.6)	
Positive	32 (51.6)	88 (62.4)	
HCV Ab			0.803
Negative	42 (67.7)	98 (69.5)	
Positive	20 (32.3)	43 (30.5)	
Liver cirrhosis			0.940
No	31 (47.7)	69 (48.3)	
Yes	34 (52.3)	74 (51.7)	
Tumor size (cm)			0.121
<5	38 (58.5)	67 (46.9)	
≥5	27 (41.5)	76 (53.1)	
Tumor number			0.963
Single	52 (80)	114 (79.7)	
Multiple	13 (20)	29 (20.3)	
Microvascular invasion			0.593
Absent	28 (49.1)	62 (44.9)	
Present	29 (50.9)	76 (55.1)	
TNM stage			0.005
I or II	47 (73.4)	75 (52.4)	
III or IV	17 (26.6)	68 (47.6)	
Histological grade			0.590
Well	14 (21.5)	27 (18.9)	
Moderate	42 (64.6)	88 (61.5)	
Poor	9 (13.9)	28 (19.6)	

Data are expressed as number (percentage). PAR2: Protease Activated Receptor-2; HBV: hepatitis B surface; HCV: hepatitis C virus; AFP: alpha-fetoprotein; TNM: tumor node metastasis.

**Table 3 medicina-57-00574-t003:** Cumulative incidence of recurrence after resection for patients with HCC.

Variable	Number of Patients	1 Year (%)	3 Years (%)	5 Years (%)	*p*-Value
Age (years)					0.062
<60	108	43.4	60.7	70.3	
≥60	100	24.3	48.1	64.3	
Gender					0.201
Female	46	26.7	49.8	64.3	
Male	162	36.8	56.2	66.3	
AFP (ng/mL)					<0.001
<200	131	22	45.4	60.9	
≥200	77	49.9	69.5	74.4	
HBsAg					0.155
Negative	88	26.7	48.7	61.4	
Positive	120	36.5	58.6	67.8	
HCV Ab					0.530
Negative	125	36	55.4	65.4	
Positive	63	24.7	54.1	65.5	
Liver cirrhosis					0.001
Absent	100	27.2	48.8	56.1	
Present	108	37	59.2	74.1	
Tumor size (cm)					0.001
<5	105	21.1	45.5	58.3	
≥5	103	43.6	63.2	73.5	
Tumor number					0.006
Single	166	30.1	51.2	61.2	
Multiple	42	40.4	65.3	81.9	
Microvascular invasion					<0.001
Absent	103	13	38	51.9	
Present	105	52	73.1	82.4	
TNM stage					<0.001
I/II	123	17.9	40.7	53	
III/IV	85	53	73.9	84.4	
Histological grade					0.001
Well	41	13.7	29.4	46.4	
Moderate	130	44	60.1	69.4	
Poor	37	47.9	65	76.7	
PAR2 expression					<0.001
Low	65	22.0	35.1	55.5	
High	143	39.7	65.3	72.7	

IHC: immunohistochemistry; WB: western blotting; HBV: hepatitis B surface; HCV: hepatitis C virus; AFP: alpha-fetoprotein.

**Table 4 medicina-57-00574-t004:** Univariate and multivariate analysis of factors associated with cumulative recurrence in HCC.

Variable	Comparison	Univariate		Multivariate	
		HR (95% CI)	*p*-Value	HR (95% CI)	*p*-Value
Age (years)	≥60 vs. <60	0.749 (0.551–1.018)	0.065		
Sex	Male vs. Female	1.272 (0.876–1.847)	0.206		
AFP (ng/mL)	≥200 vs. <200	1.675 (1.284–2.385)	<0.001	1.696 (1.196–2.403)	0.003
HBV	Positive vs. Negative	1.262 (0.913–1.746)	0.159		
HCV	Positive vs. Negative	0.896 (0.635–1.265)	0.534		
Liver cirrhosis	Presence vs. Absence	1.647 (1.204–2.253)	0.002	1.735 (1.221–2.466)	0.002
Tumor size (cm)	≥5 vs. <5	1.668 (1.228–2.266)	0.001		
Tumor number	Multiple vs. Single	1.603 (1.140–2.254)	0.007		
Microvascular invasion	Presence vs. Absence	2.625 (1.902–3.622)	<0.001		
TNM stage	III + IV vs. I + II	2.655 (1.95–3.613)	<0.001	2.061 (1.447–2.934)	<0.001
Histological grade	Poor/Moderate vs. Well	1.95 (1.304–2.916)	0.001		
PAR2 expression	High vs. Low	1.969 (1.144–3.388)	0.014	1.779 (1.181–2.681)	0.006

95% CI: 95% confidence interval; HBV: hepatitis B surface; HCV: hepatitis C virus; AFP: alpha-fetoprotein.

**Table 5 medicina-57-00574-t005:** Cumulative incidence of survival after resection in patients with HCC.

Variable	Number of Patients	1 Year (%)	3 Years (%)	5 Years (%)	*p*-Value
Age (years)					0.313
<60	108	81.7	59.8	50.9	
≥60	100	86.6	68.4	57.6	
Gender					0.382
Female	46	88.7	69.8	59.7	
Male	162	81.1	62	52.9	
AFP (ng/mL)					0.002
<200	131	88.2	72	61.4	
≥200	77	73.4	47.5	41.3	
HBsAg					0.109
Negative	88	84.1	70.3	61.7	
Positive	120	81.3	57.6	48.9	
HCV Ab					0.472
Negative	125	81.9	60.1	50.4	
Positive	63	83.3	68.1	61.7	
Liver cirrhosis					0.14
Absent	100	84	65.1	57.6	
Present	108	81.6	60.9	50.8	
Tumor size (cm)					<0.001
<5	105	94.1	74.9	62.9	
≥5	103	71.6	51.3	45.2	
Tumor no.					0.227
Single	166	84.1	66.1	56.8	
Multiple	42	77.8	51.9	44.4	
Vascular invasion					<0.001
Absent	103	95.4	82.7	72.9	
Present	105	69	42.1	33.2	
TNM stage					<0.001
I/II	123	95.7	81.1	71	
III/IV	85	64.5	37.7	30.6	
Histological grade					<0.001
Well	41	96.1	86.1	79.9	
Moderate	130	81.2	61.2	51.3	
Poor	37	73.2	43.2	34.6	
PAR2 expression					0.002
Low	65	95.2	80.9	67.3	
High	143	78.8	56.9	49.9	

HBV: hepatitis B surface; HCV: hepatitis C virus; AFP: alpha-fetoprotein; IHC, immunohistochemistry; WB, western blotting.

**Table 6 medicina-57-00574-t006:** Univariate and multivariate analysis of factors associated with overall mortality in HCC.

Variable	Comparison	Univariate		Multivariate	
		HR (95% CI)	*p*-Value	HR (95% CI)	*p*-Value
Age (Years)	≥60 vs. <60	0.938 (0.595–1.182)	0.315		
Sex	Male vs. Female	1.203 (0.793–1.826)	0.384		
AFP (ng/mL)	≥200 vs. <200	1.721 (1.221–2.426)	0.002		
HBV	Positive vs. Negative	1.346 (0.934–1.941)	0.111		
HCV	Positive vs. Negative	0.869 (0.591–1.276)	0.474		
Liver cirrhosis	Present vs. Absent	1.295 (0.917–1.828)	0.143		
Tumor size (cm)	≥5 vs. <5	2.083 (1.469–2.955)	<0.001		
Tumor number	Multiple vs. Single	1.293 (0.859–1.888)	0.229		
Microvascular invasion	Present vs. Absent	3.231 (2.122–4.696)	<0.001		
TNM stage	III + IV vs. I + II	3.356 (2.366–4.761)	<0.001	2.747 (1.851–4.077)	<0.001
Histological grade	Poor/Moderate vs. Well	2.946 (1.719–5.05)	<0.001	2.675 (1.417–5.051)	0.002
PAR2 expression	High vs. Low	2.027 (1.29 – 3.184)	0.002	1.832 (1.142 – 2.938)	0.012

## Data Availability

The data presented in this study are available on request form the corresponding authors. The data are not publicly available due to our Institutional Review Board regulation.

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
