# Peer review of "The Role of Protease-Activated Receptor 2 in Hepatocellular Carcinoma after Hepatectomy"

_medicina, 2021, doi:10.3390/medicina57060574_

Round 1

Reviewer 1 Report

In this paper, PAR2 expression  was evaluated in HCCs  and in the adjacent nontumor tissues. PAR2 overexpression was associated with advanced TNM stage and histological grade. Kaplan-Meier analysis indicated that high PAR2 expression was associated with poorer DFS and OS. Multivariate analyses indicated high PAR2 expression, liver cirrhosis, and advanced TNM  stage  were prognostic factors for DFS, and advanced TNM stage and histological grade and high PAR2 expression   were significant risk factors for OS. The combination of PAR2 expression and  serum AFP provided improved prognostic ability for OS and DFS.

This paper presents different drawbacks.

  1. The representative western analysis of only four HCCs is insufficient. For instance, the HCCs 3 and 4 showed a consistent rise in PAR2 expression, the HCC2 showed no change/small decrease and HCC1 apparently showeda  very low increase. A much higher number of HCCs musIt be analyzed.
  2. Clinicopathological features of HCC patients are largely incomplete. Some other features must be added. For instance HCC etiology (HBV, HCV, ethanol), Edmondson grade, Alpha-fetoprotein secretion, proliferation index, mean survival after partial liver resection. Also, it could be useful to evaluate an index ofinfiltrative capacity (i.e. Midkine expression) of HCCs.
  3. The survival curves are related to “High/Low” PAR2. Howeverthe definition of “high” and “low” PAR2 is lacking.
  4. Functional effects of PAR2 on tumor cell lines should be determined. Thus, the effect of PAR2 transfection onPAR2 expression, wound healing,  cell death, cell mjgration and cell invasivity should be determined.

Author Response

Reply:

Thank you for your suggestions and corrections. We explained and described step by step as below:

  1. We had added more cases (n = 7) with obviously higher PAR2 expression (n = 6) in tumor part compared with non-tumor part by WB (page 4, line 134~135).
  2. The clinicopathological data included the etiology of HCC (HBV, HCV, or other/unknown), AFP levels (median, AFP>15, and AFP>200 ng/mL), RFS and OS. About the histology grade, we recorded them as well, moderate, or poor differentiation by the World Health Organization (WHO), which was adopted in our hospital instead of Edmondson-Steiner (ES).
  3. For the definition of high or low PAR2 expression, we had defined in the page 4, line 127-129, as below: Compared with the paired nontumor tissues, high levels (defined as greater than twofold increase) of PAR2 expression in 143 of 208 (68.8%) HCC cases, and the other 65 (31.2%) were defined as PAR2 low expression group.
  4. About the function study by in vitro or vivo, we did not address more on this. In the future, we will survey the function of PAR2, such as proliferation, migration, and invasion in HCC cell lines by transfection know-down experiments. Thank you for your suggestions.

Reviewer 2 Report

It is an excellent study. Well written and very well presented. Though it has some limitations as it is retrospective, the results are significant and difficult to contest. The messages are clear. I have no any comment to make. 

Author Response

Reply:

Thank you for your comments.

Reviewer 3 Report

This paper is very interesting in the field of primary liver cancer. Correlates PAR2 levels with disease-free survival and overall survival after liver resection. The methodology is correct and the results are very well exposed.The discussion and the bibliography is adequate as well as the conclusions.

A question: there is no patient who has consumed alcohol associated or not with the viral etiology of cirrhosis?.

Author Response

Reply:

Thank you for your comments. About the alcohol consumption, we believed that there might be a certain percentage in the non-viral hepatitis group (NBNC). However, this is a retrospective study and there were no definite criteria for alcohol abuser used in our hospital. Therefore, we conserved this data in our analysis.

Round 2

Reviewer 1 Report

Functional effects of PARP2 should be determined